# Development and validation of a framework to improve neglected tropical diseases surveillance and response at sub-national levels in Kenya

Arthur K. S. Ng'etich[1]*, Kuku Voyi[1], Clifford M. Mutero[1,2,3]

**1** School of Health Systems and Public Health (SHSPH), University of Pretoria, Pretoria, South Africa, **2** University of Pretoria Institute for Sustainable Malaria Control (UP ISMC), University of Pretoria, Pretoria, South Africa, **3** International Centre of Insect Physiology and Ecology, Nairobi, Kenya

☯ These authors contributed equally to this work.
\* arthursaitabau@yahoo.com

## Abstract

### Background

Assessment of surveillance and response system functions focusing on notifiable diseases has widely been documented in literature. However, there is limited focus on diseases targeted for elimination or eradication, particularly preventive chemotherapy neglected tropical diseases (PC-NTDs). There are limited strategies to guide strengthening of surveillance and response system functions concerning PC-NTDs. The aim of this study was to develop and validate a framework to improve surveillance and response to PC-NTDs at the sub-national level in Kenya.

### Methods

A multi-phased approach using descriptive cross-sectional mixed-method designs was adopted. Phase one involved a systematic literature review of surveillance assessment studies to derive generalised recommendations. Phase two utilised primary data surveys to identify disease-specific recommendations to improve PC-NTDs surveillance. The third phase utilised a Delphi survey to assess stakeholders' consensus on feasible recommendations. The fourth phase drew critical lessons from existing conceptual frameworks. The final validated framework was based on resolutions and inputs from concerned stakeholders.

### Results

The first phase identified thirty studies that provided a combination of recommendations for improving surveillance functions. Second phase described PC-NTDs specific recommendations linked to simplified case definitions, enhanced laboratory capacity, improved reporting tools, regular feedback and supervision, enhanced training and improved system stability and flexibility. In the third phase, consensus was achieved on feasibility for implementing recommendations. Based on these recommendations, framework components constituted

**Data Availability Statement:** All relevant data are within the manuscript and its Supporting information files.

**Funding:** The author(s) received no specific funding for this work.

**Competing interests:** The authors have declared that no competing interests exist.

human, technical and organisational inputs, four process categories, ten distinct outputs, outcomes and overall impact encompassing reduced disease burden, halted disease transmission and reduced costs for implementing treatment interventions to achieve PC-NTDs control and elimination.

## Conclusion

In view of the mixed methodological approach used to develop the framework coupled with further inputs and consensus among concerned stakeholders, the validated framework is relevant for guiding decisions by policy makers to strengthen the existing surveillance and response system functions towards achieving PC-NTDs elimination.

## Author summary

Neglected tropical diseases (NTDs) affect marginalised and underserved populations with sub-national levels providing first contact healthcare services to the afflicted communities. NTDs amenable to chemoprophylaxis are primarily controlled through mass treatment interventions. However, identification of disease transmission hotspots requires strengthened health information systems (HIS) to inform targeted public health action and response. Using a multi-phased approach, we developed and validated a framework, which provided a logical approach for guiding actions to strengthen surveillance system functions in view of NTDs. Framework development involved undertaking a systematic literature review to retrieve generalised recommendations for improving surveillance system functions within the African context, conducting primary data surveys to identify disease-specific recommendations on improving surveillance system core, support and attribute functions regarding NTDs and determining feasibility for implementing recommended actions at the sub-national levels. A review of relevant conceptual frameworks provided information underpinning overall framework development. The study identified framework component interlinkages to achieve the desired results of reduced costs for implementing treatment interventions, halted disease transmission and reduced disease burden. Overall, the framework provides a logical approach for strengthening HIS at sub-national levels in NTD endemic regions, considering stakeholders' perspectives and the available resources to achieve the ultimate goal of disease elimination.

## Introduction

An integral component to achieving health systems strengthening is through enhanced performance of health information systems (HIS) [1,2]. Quality health information underpins stakeholders' actions towards achieving quality health care [3]. Essentially, well-functioning health information and surveillance systems ensure generation, analysis, distribution and utilisation of reliable information to support decision-making across all health system levels [4]. A priority action outlined in the Kenyan HIS policy framework is the integration of data collection and dissemination through partnership in health information processes involving all health service providers [2]. This closely relates to the principle of consolidated efforts within the integrated disease surveillance and response (IDSR) system framework adopted by World Health Organization (WHO) Member States in Africa [5]. The IDSR framework categorises

diseases as either being epidemic-prone, diseases of public health importance or diseases targeted for elimination or eradication [5]. In particular, neglected tropical diseases (NTDs) in Kenya are categorised as either being of public health importance or targeted for elimination or eradication [6]. NTDs are a diverse group of communicable diseases that prevail in tropical and subtropical regions, mainly affecting marginalised communities and are a major cause of deformities and disabilities [7]. The chronic nature of NTDs leads to a sustained cycle of poverty among the afflicted communities and exerts pressure on the already fragile health systems [8]. Therefore, WHO work plan for NTDs elimination identifies five intervention strategies including preventive chemotherapy among affected populations, intensified case finding (CF) and disease management, integrated vector management, improved sanitation and hygiene and veterinary public health [9]. Preventive chemotherapy (PC) is an essential strategy for NTDs infection control, which targets the disease causative agents–parasitic, viral and bacterial–concurrently through mass drug administration (MDA) [10]. However, focus on MDA strategies limits priority given to the role played by other social determinants and the existing health systems to achieve NTDs control. There is consensus among NTD experts that in order to achieve efficient and sustainable control and meet disease-specific elimination targets it requires well-functioning health systems [11–14]. Overdependence on mass treatment strategies may exhaust the available resources, thereby impeding health system strengthening efforts [11–13].

Neglected tropical diseases are a clear case in point regarding existing health systems functioning sub-optimally. NTDs are considered of low priority within national health information and disease surveillance systems, which may hinder generation of quality data and result to ineffective use and dissemination of information [15]. Dependence on chemoprophylaxis as a stand-alone intervention for PC-NTDs requires continued support of HIS components through strengthened surveillance systems. This identifies the need to establish strong and adaptable systems to improve data management capacities in NTD endemic countries [15]. Strengthened HIS will facilitate undertaking evidence-based actions and establishment of effective NTDs control interventions [11,12,15]. However, solely depending on existing interventions and technologies may not sufficiently achieve sustainable control and elimination of targeted NTDs [12,16]. Measures to strengthen existing health systems are critical to improving health personnel attitudes and credibility of the health system[17]. As a result, the need to improve present surveillance and response systems through novel frameworks tailored to local settings to inform targeted control efforts in NTD endemic areas is crucial [18].

The current framework was partly founded on the same principles adopted by conceptual frameworks for strengthening public health surveillance systems [19,20]. The intricacies of strengthening health system building blocks are evident from the varying perspectives illustrated by an array of conceptual frameworks applicable to different settings [21–23]. Frameworks developed with the intention of assessing the strength of a health system should typically address all essential components including the resources dedicated to the system (inputs), what the system inputs intend to fulfill (processes and outputs) and the resulting benefits and fundamental change to the system (outcomes and impact) [21]. However, multiple complexities are involved since the overall health system performance is dependent on several other factors involving technical, social, organizational and cultural dynamics [21,23]. The framework herein focuses on strengthening the indicator-based components of the existing surveillance system that involves the collection of structured data through routine surveillance systems [24].

Previous public health surveillance systems assessments have prioritised execution of interventions, resulting outcomes and their impact. Therefore, the proposed framework intended to conceptualise the link between inputs, processes and relevant outcomes of the existing

surveillance system. The framework was based on stakeholders' perspectives on improving surveillance and response to PC-NTDs, with the intention of strengthening implementation of surveillance activities at the county level in Kenya. The goal was to enable concerned stakeholders effectively identify gaps within the existing surveillance system, prioritise plans of actions for improving the system and design indicators for progress monitoring [25]. Therefore, the proposed framework was aimed at providing an adoptable logical approach to guide stakeholders' actions to improve surveillance and response to PC-NTDs at the sub national level in Kenya.

## Methods

### Ethics statement

Written informed consent was obtained from research participants involved at various stages of framework development and validation. Ethical approval was granted by the Faculty of Health Sciences Research Ethics Committee of the University of Pretoria in South Africa (**Ethics Reference No: 27/2018**) and the Institutional Research and Ethics Committee (IREC) of Moi University/Moi Teaching and Referral Hospital in Kenya (**Formal Approval No: IREC 2099**). In addition, the National Commission for Science, Technology and Innovation (NACOSTI) provided research authorisation to undertake the research in Kenya (**Reference No: NACOSTI/P/18/62894/21393**). Permission to conduct the overall study was granted by the Ministry of Health in Kenya and relevant County health authorities.

### Methodology for framework development

Development of the framework for improving PC-NTDs surveillance and response encompassed four main phases (Fig 1). The first phase involved conducting a systematic review of literature, which identified generalised but critical recommendations emerging from prior studies to improve disease surveillance and response systems in the African region. The second phase involved primary data surveys conducted in PC-NTD endemic regions in Kenya to assess performance of surveillance core, support and attribute functions. The third phase used a modified-Delphi survey to assess stakeholders' consensus on the importance and feasibility of implementing recommended actions to improve PC-NTD surveillance and response at the sub-national level in Kenya. Lastly, the fourth phase involved a review of previous HIS and public health surveillance system conceptual frameworks to gain insights into the existing frameworks.

### Phase I: Systematic review of literature on surveillance system assessment studies [26]

The purpose for this phase was to systematically review both published and grey literature retrieved from the relevant online databases including PubMed, Web of Science, Scopus, Cumulative Index for Nursing and Allied Health Literature (CINAHL) and World Health Organisation Library and Information Networks for Knowledge (WHOLIS). The focus of the systematic literature review process was to search for studies assessing disease surveillance and response systems in the African region based on health workers' perspectives (PROSPERO Registration number CRD42019124108) [26]. The review focused on the post-adoption phase of the revised IDSR system as from 2010 and onwards. A combination of keywords were used to search for relevant studies including "surveillance", "public health surveillance", "integrated disease surveillance and response", "evaluation", "assessment", "health worker", "healthcare personnel", "Africa" and "Sub Saharan Africa". The review aimed to assess key findings and

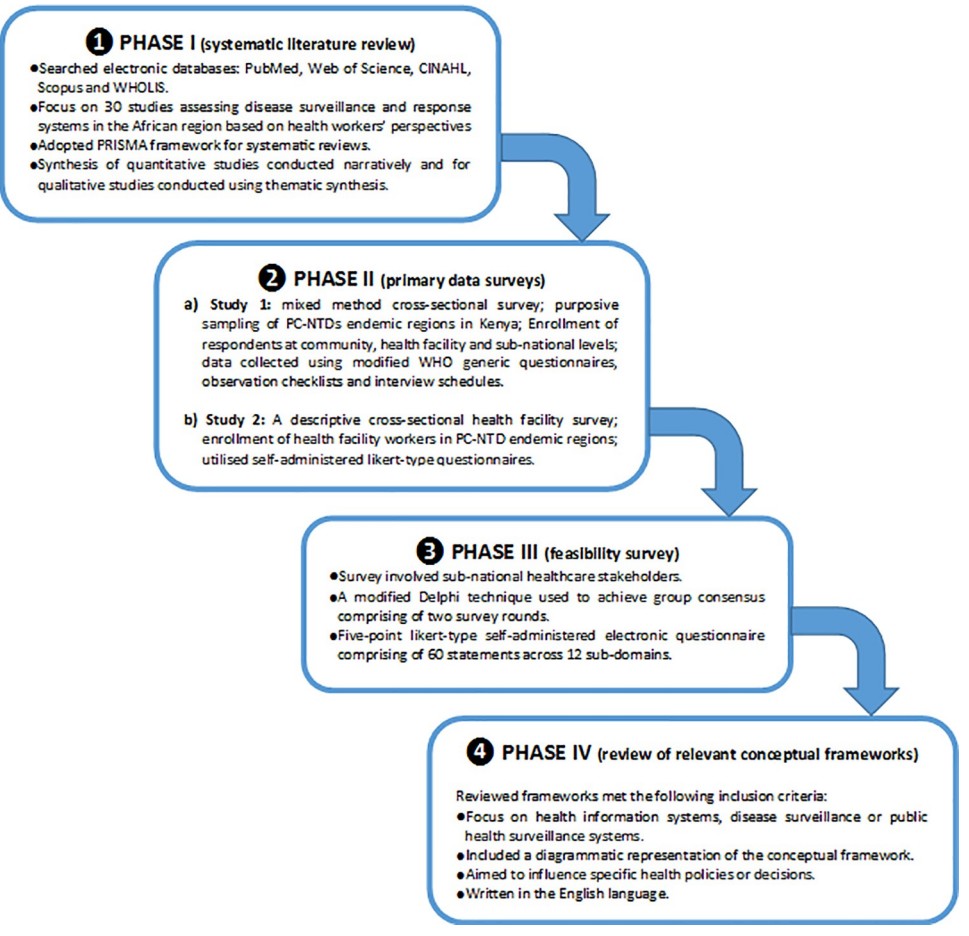

**Fig 1. Framework development process.**

recommendations to improve surveillance and response systems in the African region based on health workers' perspectives.

## Phase II: Primary data surveys on surveillance core, support and attribute functions concerning PC-NTDs [27,28]

The aim of this phase was to conduct two empirical studies in NTDs endemic regions in Kenya. The studies aimed to assess performance of surveillance functions concerning PC-NTDs as perceived by healthcare workers. The two studies intended to put focus on improving PC-NTDs surveillance within the existing IDSR system, which is in line with the strategic objectives outlined in the Second National Strategic Plan for Neglected Tropical Diseases and the National Breaking Transmission Strategy in Kenya [29,30]. The first study used a mixed method cross-sectional survey to assess surveillance system core and support functions relating to PC-NTDs [27]. Core functions were based upon indicators that measured the system processes and outputs including case detection, registration and confirmation of health related events, reporting, analysis and interpretation of surveillance data and public health action including reporting and feedback to users of the system. On the other hand, support functions facilitated implementation of surveillance core functions comprising of standards and guidelines, training, supervisory activities, communication, and human, technical and

financial resources [31]. The study sites comprised of 10 purposively sampled counties from three distinct geographical regions in Kenya including Rift Valley (Baringo, Narok and West Pokot), Coast (Kwale, Kilifi, Lamu, Tana River and Taita Taveta) and Eastern (Kitui and Embu) regions, which are co-endemic of at least three or more fully mapped PC-NTDs. The health system in Kenya is organised into four distinct levels: Tier I, also referred to as community health services; Tier II, identified as primary care level comprising of dispensaries and sub-county healthcare facilities; Tier III comprising of county referral hospitals; and Tier IV that encompasses all national referral hospitals. The sub-national levels constitute county and sub county levels including level 5, level 4 and level 3 facilities while the peripheral levels comprise of level 2 health facilities and the community health units. Health personnel involved in the study were enrolled from the community (50), health facility (192) and sub-national (44) levels. Modified WHO generic questionnaires, observation checklists and interview schedules were used for data collection. Semi-structured questionnaires were used to obtain information from healthcare personnel responsible for surveillance data collection and transmission at the community, healthcare facility, sub-county and county levels. In addition, interview schedules were used to conduct key informant interviews with healthcare personnel overseeing disease surveillance related activities in the PC-NTD endemic counties (S1 File). Both quantitative and qualitative data were concurrently analysed and the findings were integrated. The second study aimed to evaluate surveillance system attributes based on health facility workers' perceptions concerning PC-NTDs [28]. Surveillance attribute functions included: (i) simplicity (ease of operation of the surveillance system); (ii) acceptability (willingness of persons to be involved in activities within the surveillance system); (iii) stability (ability of surveillance system to be available and reliable when required); (iv) flexibility (ease of surveillance system to adapt to change of information needs and operating conditions with minimal additional resources); (v) usefulness (surveillance system contribution to control and prevention of adverse health-related conditions); (vi) data quality (accuracy of data collected within the surveillance system), (vii) surveillance data timeliness (how quick information is conveyed across levels of the surveillance system); and, (viii) completeness (expected essential data requirements compared to actual reporting) [31]. The study sites were similar to those selected in the first survey. The study adopted a descriptive cross-sectional health facility survey involving 192 health workers responsible for disease surveillance and response activities in NTD endemic regions in Kenya. Self-administered likert-type questionnaires were used to assess healthcare personnel perceptions of surveillance system attributes on simplicity, acceptability, stability, flexibility, usefulness, data quality and reporting timeliness and completeness with regard to PC-NTDs.

## Phase III: Feasibility of implementing recommendations to improve PC-NTDs surveillance and response [32]

This third phase was based on a Delphi survey, which involved enrollment of key stakeholders at the sub-national level responsible for overseeing disease surveillance and response activities in Kenya [32]. The Delphi procedure is a scientific method that uses a series of rounds to gather individual expert opinions in order to reach a consensus among participants. To obtain group consensus, the structured approach uses a combination of questionnaires or iterative rounds to collect data [33,34]. The survey aimed to assess the principal and feasible recommendations derived from the preceding first and second phase. This phase guided selection of key constructs making up the framework in line with stakeholders' opinions and inputs. Stakeholders constituted individuals that regularly utilise information generated by the surveillance system to take appropriate actions or make decisions influencing implementation of interventions and policy-making processes [35,36].

### Phase IV: Review of relevant conceptual frameworks

In the fourth phase, a comprehensive review of existing HIS and public health surveillance system conceptual frameworks was undertaken. Retrieval of the relevant documents involved basic search strategies; electronic database search, a manual reference search of selected documents and use of a generic internet search engine. Literature searches were undertaken using PubMed, Web of Science, Scopus, Google Scholar and a free Google search to identify the relevant frameworks. The key search terms included: "conceptual framework", "model", "health systems", "health information systems", "public health surveillance systems", "surveillance and response". These terms were used in various combinations to ensure the literature search was extensive. A manual reference check of the included frameworks was conducted to identify additional frameworks that met the inclusion criteria. The inclusion criteria for the reviewed frameworks was a specific focus on HIS, disease surveillance or public health surveillance systems; included a diagrammatic representation of the conceptual framework; aimed to influence specific health policies or decisions; and written in the English language. This review involved critical assessment of components making up the frameworks. In addition, reviewing the framework components provided an understanding on the relevant interlinkages, which formed the basis for the proposed framework.

### Data management and analysis

In the first phase, data synthesis for quantitative studies was conducted narratively while analysis of extracted data from qualitative studies was done using thematic synthesis. A matrix of main themes of surveillance functions guided the thematic synthesis with emerging sub-themes. In the second phase, analysis of data in the first study involved frequency computations and summaries of respondents' socio-demographic characteristics, with bivariate analysis between categorical variables assessed using Pearson's Chi Square test. Independent variables associated with the outcome in the bivariate analysis were incorporated in mixed effects logistic regression models. Qualitative data analysis identified the main themes based on a coding framework. In the second study, frequency distributions for each question in the five-point likert scale were analysed, while reporting rates were summarised using percentages and estimated median values. Internal consistency for the group of variables in each attribute domain was assessed using Cronbach's Alpha. In the third phase, categorical variables were summarised using descriptive statistics while participant responses measured on a likert scale were reported using frequency distributions. Consensus was defined as >70% of participants either "agreeing" or "strongly agreeing" with the recommendation statements in each of the two rounds. Reliability and coherence between items in questionnaires used in both rounds were independently tested using Cronbach's Alpha coefficient. In the final phase, narrative synthesis was utilised by thematically coding lessons drawn from the existing conceptual frameworks.

## Results

### Phase I: Systematic review of literature on surveillance system assessment studies [26]

Thirty studies met the inclusion criteria and were assessed to retrieve generalised recommendations to improve surveillance systems within the revitalised IDSR framework adopted by African countries. Eighteen emerging sub-themes were identified for recommendations specific to improving four core surveillance functions and three support functions. However, no specific sub-themes emerged from assessing the surveillance system attributes. The

surveillance core functions included; case confirmation, reporting, data analysis and feedback. Surveillance support functions included; training, supervision and resources while surveillance system attributes included; reporting timeliness and completeness, data quality and accuracy, usefulness, acceptability, simplicity and stability. Other key recommendations alluded to adoption of alternative surveillance strategies and calls for further research to strengthen surveillance systems.

Main themes were based on elements constituting the broader surveillance functions–core, support and attributes–while 18 emerging sub-themes were derived from the recommendations [37–64] (Fig 2). Other recommendations included; alternative surveillance strategies (electronic based surveillance [39,48,53,54,60,65], community based surveillance [45,61] and syndromic surveillance [65]) and further calls for research on surveillance [46,54,62].

## Phase II: Primary data surveys on surveillance core, support and attribute functions concerning PC-NTDs [27,28]

This first study identified specific recommendations to strengthen surveillance functions focusing on PC-NTDs within the existing IDSR system. There were eleven a priori identified main themes with up to 62 emerging sub-themes based on recommendations to improve PC-NTDs surveillance and response according to health workers' perspectives. A high degree of groundedness was defined as sub-themes (recommendations) that were mentioned fifteen or more times ($G \geq 15$) by the research participants under each main theme (S1 and S2 Tables). Recommendations to improve PC-NTDs surveillance core activities were categorised into 7 main themes and 32 sub-themes. The principal recommendations considering a high degree of groundedness included provision of simplified case definitions, improved laboratory capacity, provision of PC-NTDs reporting tools, timely and regular feedback on surveillance reports, adoption of electronic feedback mechanisms, improved data analysis skills and provision of adequate outbreak response supplies. On the other hand, recommendations to strengthen surveillance support activities comprised of 4 main themes and 30 sub-themes. Therefore, the main recommendations to improve support functions based on a high degree of groundedness encompassed provision of surveillance guidelines for supervision, properly constituted supervisory teams, regular supervision from higher levels, frequent provision of PC-NTDs updates, regular sensitisation and involvement of all health workers in surveillance activities, enhanced training on PC-NTDs surveillance, improved human resource capacity, provision of surveillance reporting and analysis tools, availing equipment and training materials, provision of adequate funding to facilitate surveillance activities and provision of reliable transportation.

The subsequent study showed that health workers perceived the surveillance system to be simple (55%), acceptable (50%), stable (41%), flexible (41%), useful (51%) and to provide quality surveillance data (25%). Health personnel experienced difficulties completing IDSR reporting forms with regard to PC-NTDs, easily understanding reporting guidelines and utilising PC-NTDs case definitions. Further, health workers were less satisfied with their involvement in facility-based surveillance activities. Health workers perceived PC-NTDs to be of low priority compared to other conditions and were only willing to be involved in surveillance activities of diseases considered of priority. However, community level health workers were more willing to support and be involved in PC-NTDs surveillance activities. Heath workers felt the existing IDSR system lacked flexibility with regard to being well adapted to report all PC-NTDs co-endemic in the region and easily adapting to changes in information needs, technological shift and funding. Furthermore, minimal adaptability of the existing surveillance system to changes in reporting mechanisms from paper-based to electronic systems was reported.

**Core functions**

- **Case confirmation** (*Improved specimen handling; Strengthened laboratory support*)
- **Reporting** (*Improved reporting quality; Adequate reporting forms provision*)
- **Data analysis** (*Increased surveillance performance monitoring; Improved data accuracy*)
- **Feedback** (*Improved health workers' attitudes; Enhanced feedback from higher to lower levels*)

**Support functions**

- **Supervision** (*Strengthened implementation of surveillance systems; Utilisation of up-to-date information; Identification of correct reporting channels*)
- **Training** (*Improved performance of the surveillance system; Improved surveillance data quality; Enhanced knowledge on surveillance systems*)
- **Resources** (*Financial resources; Human resources; Technical, material and logistical resources; Equipment and infrastructure*)

**Surveillance attributes**

- **Simplicity**
- **Acceptability**
- **Stability**
- **Flexibility**
- **Usefulness**
- **Data quality and accuracy**
- **Timeliness and completeness**

**Main themes**

*(Emerging themes)*

**Fig 2. Surveillance system functions (main themes and emerging sub-themes).**

Respondents further indicated low stability of the surveillance system in terms of adequacy of the available forms to report PC-NTDs, resource sufficiency and capacity of health managers to support surveillance activities and address challenges promptly. Health workers also reported that the surveillance system lacked usefulness in terms of generating sufficient PC-NTDs surveillance data to inform implementation of control interventions and influence sufficient donor support. In addition, respondents felt the surveillance system lacked capacity to provide adequate information to inform efficient public health actions in view of PC-NTDs. Health facilities sampled from the PC-NTDs endemic regions hardly met the 80% target reporting thresholds. Only about one-third of the facilities met this threshold in terms of reporting timeliness of monthly surveillance data and slightly more than a half of facilities met the threshold for completeness of monthly reports. Overall, considering a cut off attribute score of above 50%, findings showed health workers perceived the existing surveillance system to be simple, acceptable and useful with regard to PC-NTDs. However, respondents had low perceptions on the stability, flexibility and data quality of the surveillance system. Notably, there was a dwindling trend in monthly reporting timeliness and completeness rates by health facilities over a three-year period.

## Phase III: Feasibility of implementing recommendations to improve PC-NTDs surveillance and response [32]

Findings from the Delphi survey indicated that stakeholders agreed on the importance of 56 (93%) recommendation statements and further reached consensus on the feasibility of implementing 47 (84%) recommended priority actions at the sub-national level. Stakeholders had a converging opinion on the importance and feasibility of implementing recommendations in six sub-domains relating to feedback, epidemic preparedness and response and those relating to four surveillance system attributes–simplicity, acceptability, stability and flexibility. However, there was lack of consensus on specific recommendations regarding the remaining six sub-domains on their importance or feasibility for implementation. Furthermore, sensitivity analysis that incorporated those participants with neutral responses indicated that the participants considered all recommendation statements to be of importance. Nevertheless, there was still non-consensus on the feasibility of availing disease-specific case registers, confirmation of all PC-NTDs cases, undertaking routine data analysis, increasing the number of supervisory visits at the lower levels, involving all health workers in surveillance training, retaining trained surveillance personnel and increasing the number of health workers responsible for overseeing surveillance activities at the sub-national level.

## Phase IV: Review of relevant conceptual frameworks

The systematic search identified 10 records eligible for full-text review. However, only 6 distinct conceptual frameworks focusing on HIS and public health surveillance systems met the inclusion criteria for review (S3 Table). Critical lessons drawn from HIS conceptual frameworks alluded to stakeholders' involvement, the organisational setting and technological aspects. Further, reviewed public health surveillance frameworks focused on performance indicators, disease-specific surveillance aspects and outbreak detection. Lessons learned from the reviewed conceptual frameworks alluded to stakeholders' involvement, framework components and their interlinkage and disease-focused perspectives [19–23,36].

## Development of a framework to improve PC-NTDs surveillance and response

Framework development was informed by feasible recommendations derived from the third phase; in addition, to the lessons drawn from reviewing relevant conceptual frameworks. The

proposed framework aimed to provide a logical approach hinged on the inputs, processes and outputs to enable decision makers institute evidence-based actions to achieve the intended outcomes and the desired impact [66]. A stakeholder-oriented approach was adopted throughout the framework development process to identify practically feasible recommendations for implementation and ensure utilisation of evidence-based research findings for decision-making [67]. The underlying assumption was that research findings can only be reliable to effectively influence the policy making process if they are endorsed by concerned stakeholders [67]. This assumption formed the basis for undertaking the third phase, which identified key components of the proposed framework. We also adopted the logical framework approach (LFA) as stipulated in the guide to monitoring and evaluating communicable disease surveillance and response systems [68]. The framework components were categorised based on their focus on input factors (resources), process factors (planned activities), output factors (short, medium and long-term benefits), outcome factors (intended results) and the overall impact factors [68]. Framework development was guided by findings from the first, second, third and fourth phases with the aid of a logical component matrix to identify the interrelation between the derived themes (Table 1). The proposed framework intended to strengthen implementation of the core medical interventions for PC-NTDs control; preventive chemotherapy and intensified CF and disease management [9,69].

The first step to developing the proposed framework involved analysing the required inputs. The existing surveillance and response system requires resource mobilisation through injection of inputs to facilitate the planned work in view of NTDs amenable to chemoprophylaxis at the sub-national levels. Therefore, stakeholders' consensus on the feasible inputs were categorised into two sub-domains with the first being a combination of surveillance tools, equipment and infrastructure including; data analysis tools and equipment such as computers with pre-loaded analysis software, clearly formulated action thresholds, outbreak preparedness and response protocols and guidelines, surveillance activities supervisory schedules, training materials and electronic communication devices. The second sub-domain combined financial, technical and logistical support including; increased funding allocation to support surveillance activities, enhanced training on data analysis, provision of sufficient emergency supplies for outbreak response, reliable transportation and other logistical support.

The second step involved identification of practically feasible processes. The activities were categorised into four sub-domains; (i) strengthening existing surveillance tools, (ii) surveillance core activities, (iii) surveillance support activities, and (iv) surveillance system attributes concerning PC-NTDs. Actionable processes for strengthening the existing surveillance tools included; updating surveillance guidelines and the available standard case definitions currently in use and inclusion of all PC-NTDs in the reporting forms. Processes to strengthen the core activities in view of PC-NTDs included; fully equipping health facility level laboratories, availing updated reporting guidelines, ample time allocation for surveillance reports preparation and submission, enhanced PC-NTDs data analysis, provision of regular and timely feedback, adoption of electronic feedback mechanisms, increased feedback to lower levels and properly constituting outbreak response teams. Additionally, activities to strengthen support activities included; prioritising supervision of PC-NTDs surveillance activities, adequately constituting the supervisory teams, increasing community level involvement in supervisory activities, prioritising and conducting regular PC-NTDs surveillance training, improving telecommunication channels for data transmission and improving access to reliable transportation when undertaking surveillance activities. Lastly, processes to strengthen surveillance attributes regarding PC-NTDs included; simplification of existing reporting guidelines, reporting forms, standard case definitions, and data collection and analysis procedures, increased support for surveillance activities by health managers, consideration of PC-NTDs as conditions of public

**Table 1. Logical framework component matrix.**

| Phase II: Primary data studies Logical framework components *(Paraphrased statements)* | Component III: Output themes | Component IV: Outcome themes | | | Phase I: Systematic literature review themes | Phase III: Feasible actions (Yes/No) | Phase IV: Reviewed conceptual frameworks themes |
|---|---|---|---|---|---|---|---|
| | | Early detection and effective response action | Improved data accuracy and quality | Efficient surveillance data transmission | | | |
| **Component I: Inputs** | | | | | | | |
| **Input I: Resources–tools, equipment and infrastructure** | | | | | | | |
| Availing case registers* All diseases are recorded in a common register due to lack of specific case registers for registration of NTDs presenting at the facilities Provision of disease registers specific for NTDs across all surveillance levels requires additional resources | Accurate case registration and reporting | √ | √ | √ | - | No | Framework component (Input) |
| Increased laboratories at lower surveillance levels* Supervisory team recommend specimen collection and forwarding to sub-county level for confirmation | Improved case confirmation capacity | √ | - | - | Strengthened laboratory support | No | Framework component (Input) |
| Adoption of electronic reporting tools Having adopted electronic reporting systems in the previous years has enabled timely data transmission Having an electronic reporting system at the facility level would ease reporting and eliminate the burden of physically submitting surveillance reports | Accurate case registration and reporting | √ | - | - | Improved reporting | Yes | Framework component (Input) |
| Clearly formulated action thresholds Action thresholds guide health workers on when to implement certain measures such as school-based deworming exercises Preset action thresholds for NTDs are hardly utilised with priority given to notifiable conditions NTDs action thresholds are not well understood among health workers due to limited knowledge on data analysis | Improved epidemic preparedness and response | √ | - | - | - | Yes | Framework component (Input) |
| Adequate data analysis tools and equipment Health workers need to be provided with sufficient guidelines to motivate conducting data analysis frequently Lack of appropriate data analysis tools affects the frequency of conducting analysis of surveillance data collected | Strengthened data analysis | √ | √ | - | Improved data accuracy | Yes | Framework component (Input) |

*(Continued)*

**Table 1.** (Continued)

| Phase II: Primary data studies Logical framework components *(Paraphrased statements)* | Component III: Output themes | Component IV: Outcome themes | | | Phase I: Systematic literature review themes | Phase III: Feasible actions (Yes/No) | Phase IV: Reviewed conceptual frameworks themes |
|---|---|---|---|---|---|---|---|
| Provision of updated outbreak preparedness and response protocols *Preparedness for epidemics can be improved by providing the appropriate outbreak preparedness plans and manuals* | Improved epidemic preparedness and response | √ | - | - | - | Yes | Framework component (Input) |
| Formulation of supervisory schedules for surveillance activities *We require a formal supervision plan specific for monitoring surveillance activities throughout the year Lower levels lacked specific schedules for supervision and only conduct visits whenever a need arises* | Enhanced supervision of surveillance activities | - | √ | √ | Strengthened implementation of surveillance system activities | Yes | Framework component (Input) |
| Availing adequate training materials and equipment across all surveillance levels *Low priority is given to distributing training materials to improve capacity of health workers on PC-NTDs surveillance Training materials tend to be sufficiently provided at higher level facilities but with limited provision at the lower facility levels* | Enhanced resource capacity to support surveillance activities | √ | √ | √ | - | Yes | Framework component (Input) |
| Availing electronic communication equipment for transmission of surveillance data *Electronic communication mechanisms will ensure feedback is relayed in a timely manner Inadequate and non-functional electronic equipment limit transmission of surveillance reports* | Improved training coverage on surveillance activities Enhanced resource capacity to support surveillance activities | √ √ | √ √ | √ √ | Improved surveillance data quality | Yes | Framework component (Input) Components interlinkage |
| **Input II: Resources—financial, technical and logistical support** | | | | | | | |
| Provision of an adequate number of skilled laboratory personnel *Community levels lack capacity to confirm suspected PC-NTD cases due to the unavailability of trained laboratory personnel NTD endemic regions have limited number of well-trained laboratory staff with skills for case confirmation* | Improved case confirmation capacity | √ | - | - | Improved specimen handling | Yes | Framework component (Input) |
| Enhanced training on surveillance data analysis *Lack of skills and knowledge to adequately conduct data analysis of surveillance data. Further training on surveillance data analysis is required since most health workers only have the basic skills for analysis of surveillance* | Strengthened data analysis Improved training coverage on surveillance activities | √ √ | √ √ | - √ | Improved surveillance data quality | Yes | Framework component (Input) Components interlinkage |

*(Continued)*

**Table 1.** (Continued)

| Phase II: Primary data studies Logical framework components *(Paraphrased statements)* | Component III: Output themes | Component IV: Outcome themes | | | Phase I: Systematic literature review themes | Phase III: Feasible actions (Yes/No) | Phase IV: Reviewed conceptual frameworks themes |
|---|---|---|---|---|---|---|---|
| Emergency supplies to respond to probable outbreaks<br>*Lack of adequate supplies pose a challenge to health workers in terms of responding to outbreaks*<br>*More resource support in terms of supplies is required for adequate response to PC-NTDs epidemics* | Improved epidemic preparedness and response Enhanced resource capacity to support surveillance activities | √<br>√ | -<br>√ | -<br>√ | - | Yes | Framework component (Input) |
| Increased funding to support PC-NTDs surveillance activities<br>*Availing funds to facilitate the trainings for health workers*<br>*Budget allocations hardly factor in funds specific for conducting NTDs surveillance activities*<br>*Provision of adequate funds to facilitate conducting the supervisory visits at the lower levels*<br>*Lack of funds to adequately cover transportation costs* | Enhanced resource capacity to support surveillance activities | √ | √ | √ | - | Yes | Framework component (Input) |
| Improved transport and logistical support to facilitate surveillance activities<br>*Lack of adequate resources to transport specimen to the next level for confirmation*<br>*Unreliable means of transport affects timely transfer of NTD samples from the lower levels* | Enhanced resource capacity to support surveillance activities | √ | √ | √ | - | Yes | Framework component (Input) |
| Increasing the number of health workers involved in surveillance activities*<br>*Having health staff designated to handle surveillance activities and reports submission at the facility level* | Enhanced resource capacity to support surveillance activities | √ | √ | √ | - | No | Framework component (Input) |
| Component II: Process | | | | | | | |
| Process I: Strengthening existing surveillance tools | | | | | | | |
| Update surveillance guidelines currently in use<br>*Lack of up-to-date guidelines to efficiently identify and report PC-NTDs cases*<br>*Improving data analysis skills through provision of updated guidelines and training manuals for conducting analysis of surveillance data*<br>*Laboratory personnel require up-to-date case confirmation guidelines* | Accurate case registration and reporting Improved case confirmation capacity Strengthened data analysis | √<br>√<br>√ | √<br>-<br>√ | √<br>-<br>- | Improved reporting Improved data accuracy | Yes | Framework component (Process) Components interlinkage |

*(Continued)*

**Table 1.** (Continued)

| Phase II: Primary data studies Logical framework components (Paraphrased statements) | Component III: Output themes | Component IV: Outcome themes | | | Phase I: Systematic literature review themes | Phase III: Feasible actions (Yes/No) | Phase IV: Reviewed conceptual frameworks themes |
|---|---|---|---|---|---|---|---|
| Update the available PC-NTDs case definitions currently in use *Provision of updated standard case definition guidelines for effective case identification and notification Need for clear visual aids and posters for NTDs standard case definitions to be displayed at the health facility level* | Accurate case registration and reporting | √ | √ | √ | - | Yes | Framework component (Process) |
| Listing all PC-NTDs in the existing reporting forms *NTDs are still neglected in the available reporting form; not listed in the summary forms signifying lack of priority Reporting NTDs as "other" conditions creates a low perception on their importance and influences health workers' reporting attitudes* | Accurate case registration and reporting Prioritisation of PC-NTDs surveillance activities | √ √ | √ √ | √ √ | Adequacy of reporting forms provision | Yes | Framework component (Process) |
| Process II: Strengthening existing surveillance core activities | | | | | | | |
| Confirmation of all PC-NTDs at the lower surveillance levels* *Peripheral level facilities lack laboratories and have inadequate equipment for NTDs specimen collection, storage and transportation* | Improved case confirmation capacity | √ | - | - | - | No | Framework component (Process) |
| Properly equipping laboratories at the health facility level to improve PC-NTDs case confirmation *Require fully equipped laboratories with trained laboratory personnel at the lower levels Facilities lack updated case confirmation guidelines and adequate laboratory reagents to confirm PC-NTDs* | Improved case confirmation capacity | √ | - | - | - | Yes | Framework component (Process) |
| Ensuring reporting forms are always readily available across all surveillance levels *Always having reporting forms available at the facility level will ensure timely reporting Timely transmission of surveillance data is dependent on the availability of reporting forms* | Accurate case registration and reporting | √ | √ | √ | - | Yes | Framework component (Process) |

(*Continued*)

**Table 1.** (Continued)

| Phase II: Primary data studies Logical framework components *(Paraphrased statements)* | Component III: Output themes | Component IV: Outcome themes | | | Phase I: Systematic literature review themes | Phase III: Feasible actions (Yes/No) | Phase IV: Reviewed conceptual frameworks themes |
|---|---|---|---|---|---|---|---|
| Availing updated reporting guidelines *Surveillance forms need to have specific provision for NTDs as opposed to reporting them as other conditions* *Need to update IDSR guidelines and reporting forms for capturing NTDs that are prevalent in the region* *Accurate capture of NTDs requires modified reporting forms* | Accurate case registration and reporting | √ | √ | √ | Improved reporting Identification of correct reporting channels | Yes | Framework component (Process) |
| Immediate reporting of PC-NTD cases* *Lack of reporting tools makes notification of NTDs cases challenging* *Improved communication channels at the peripheral levels would improve notification of suspected PC-NTD cases* | Accurate case registration and reporting Prioritisation of PC-NTDs surveillance activities Improved epidemic preparedness and response | √ √ √ | √ √ - | √ √ - | Identification of correct reporting channels | No | Framework component (Process) |
| Allocate adequate time for surveillance reports preparation and submission to the next levels *Preparing surveillance reports becomes time-consuming especially with other competing tasks* | Accurate case registration and reporting | √ | √ | √ | Improved reporting | Yes | Framework component (Process) |
| Prioritising PC-NTDs data analysis *Deployment of health workers knowledgeable on data analysis and management* *Increased sensitisation and regular training of health workers on conducting data analysis* | Strengthened data analysis Prioritisation of PC-NTDs surveillance activities | √ √ | √ √ | - √ | Improved data accuracy | Yes | Framework component (Process) Components interlinkage |
| Analysis of PC-NTDs data on a routine-basis* *Data analysis is not routinely done because health workers lack the adequate skills* *Availing the relevant electronic equipment will facilitate conducting routine data analysis* | Strengthened data analysis | √ | √ | - | Surveillance system performance monitoring | No | Framework component (Process) |
| Periodic trend analysis of PC-NTDs reported cases* *Frequent data analysis will enable assessment of disease trends* *Surveillance data collected is not sufficient to conduct trend analysis* | Strengthened data analysis | √ | √ | - | Surveillance system performance monitoring | No | Framework component (Process) |

*(Continued)*

**Table 1.** (Continued)

| Phase II: Primary data studies Logical framework components *(Paraphrased statements)* | Component III: Output themes | Component IV: Outcome themes | | | Phase I: Systematic literature review themes | Phase III: Feasible actions (Yes/No) | Phase IV: Reviewed conceptual frameworks themes |
|---|---|---|---|---|---|---|---|
| Prioritising feedback regarding PC-NTDs surveillance data *Frequent feedback on reports submitted to higher levels will motivate health workers to improve reporting* *Written feedback enables effective referencing and ensures health workers use the reports as guidelines for taking action* | Improved feedback on surveillance data Prioritisation of PC-NTDs surveillance activities | - √ | √ √ | - √ | Improved health workers' attitudes | Yes | Framework component (Process) |
| Provision of timely and regular feedback on reported PC-NTDs surveillance data *Timely and complete feedback improves health worker reporting performance* *Receiving regular feedback on submitted reports shows the health workers that their reports are valued and appreciated* | Improved feedback on surveillance data | - | √ | - | Improved health workers' attitudes | Yes | Framework component (Process) |
| Adoption of improved electronic feedback mechanisms *Need to improve feedback mechanisms by adapting electronic methods to ensure timely feedback* *The existing mechanisms for relaying feedback on surveillance reports to the lower levels need improvement* | Improved feedback on surveillance data Enhanced resource capacity to support surveillance activities | - √ | √ √ | - √ | Enhanced feedback from higher to lower surveillance levels | Yes | Framework component (Process) Components interlinkage |
| Increased feedback on PC-NTDs to lower surveillance levels *Feedback reports from higher levels should be relevant to lower surveillance levels; feedback needs to be specific to surveillance activities undertaken at the facility generating the reports* *Lack of reliable feedback mechanisms to the lower levels* | Improved feedback on surveillance data | - | √ | - | Enhanced feedback from higher to lower surveillance levels | Yes | Framework component (Process) |
| Well-constituted outbreak response teams to respond to probable PC-NTDs outbreaks *Having properly constituted outbreak response teams in the region will improve control and response to NTD epidemics* | Improved epidemic preparedness and response Prioritisation of PC-NTDs surveillance activities | √ √ | - √ | - √ | - | Yes | Framework component (Process) |
| Process III: Strengthening existing surveillance support activities | | | | | | | |

*(Continued)*

**Table 1.** (Continued)

| Phase II: Primary data studies Logical framework components (Paraphrased statements) | Component III: Output themes | Component IV: Outcome themes | | | Phase I: Systematic literature review themes | Phase III: Feasible actions (Yes/No) | Phase IV: Reviewed conceptual frameworks themes |
|---|---|---|---|---|---|---|---|
| Prioritising supervision of PC-NTDs surveillance activities *The agenda of supervisory visits are hardly specific to PC-NTDs surveillance activities; priority given to notifiable conditions* | Enhanced supervision of surveillance activities Prioritisation of PC-NTDs surveillance activities | - √ | √ √ | √ √ | Strengthened implementation of surveillance system activities | Yes | Framework component (Process) |
| Regular supervision of PC-NTDs surveillance activities at the lower levels* *Increasing the frequency of supervision of surveillance activities especially at the lower levels* *Regular supervisory visits at lower levels to monitor use of updated guidelines and surveillance tools* | Enhanced supervision of surveillance activities | - | √ | √ | Utilisation of up-to-date information | No | Framework component (Process) |
| Properly constituting supervisory teams to adequately supervise PC-NTDs surveillance activities *Supervisory teams to always comprise of surveillance focal persons* *Supervisory teams need to be knowledgeable on PC-NTDs to accord them priority* | Enhanced supervision of surveillance activities Prioritisation of PC-NTDs surveillance activities | - √ | √ √ | √ √ | - | Yes | Framework component (Process) |
| Increased community level participation to support supervision of PC-NTDs surveillance activities *Lack of functional community health units and demoralised community-based health workers sometimes unwilling to get involved in surveillance activities* *Community health units are fundamental to having a functional surveillance system at the community level* *Regular supervision at the lower levels will improve surveillance activities being conducted at this levels and enhance active case finding* | Enhanced supervision of surveillance activities | - | √ | √ | Strengthened implementation of surveillance system activities | Yes | Framework component (Process) |
| Prioritising PC-NTDs in surveillance training *Health worker sensitisation on surveillance data analysis in PC-NTD endemic regions* *There is rarely any formal training specific to NTDs or their surveillance activities* *Training needs assessment for NTD surveillance should be conducted frequently* *Need for increased resource support for post-basic training on NTDs* | Improved training coverage on surveillance activities Prioritisation of PC-NTDs surveillance activities | √ √ | √ √ | √ √ | Improved surveillance data quality | Yes | Framework component (Process) |

*(Continued)*

**Table 1.** (Continued)

| Phase II: Primary data studies Logical framework components *(Paraphrased statements)* | Component III: Output themes | Component IV: Outcome themes | | | Phase I: Systematic literature review themes | Phase III: Feasible actions (Yes/No) | Phase IV: Reviewed conceptual frameworks themes |
|---|---|---|---|---|---|---|---|
| Regular training on PC-NTDs surveillance activities *Frequent training enables health workers to easily and consistently apply the available case definitions for NTDs* *More training and awareness among health workers on conducting analysis of NTDs surveillance data is required* *Regular training will ensure that supervisory visits are conducted in a standardised manner reviewing all disease surveillance activities* *Important to have on-job trainings and sensitisation of health workers on supervisory activities at the lower levels* | Accurate case registration and reporting Enhanced supervision of surveillance activities Improved training coverage on surveillance activities | √ - √ | √ √ √ | √ √ √ | Improved surveillance system performance Enhanced knowledge on surveillance systems Utilisation of up-to-date information | Yes | Framework component (Process) Components interlinkage |
| Involvement of all health workers in training on PC-NTDs surveillance activities *Not much priority is given to training of lower level health workers coupled by minimal effort by trained staff to cascade knowledge to other staff at peripheral levels* *Involvement of all health workers in surveillance training ensures their needs are met and they receive the relevant up-to-date information* | Improved training coverage on surveillance activities | √ | √ | √ | Enhanced knowledge on surveillance systems | No | Framework component (Process) Stakeholders' involvement |
| Retaining trained surveillance staff across all surveillance levels* *Retention of trained staff ensures consistency in conducting surveillance activities* | Improved training coverage on surveillance activities | √ | √ | √ | Improved surveillance system performance | No | Framework component (Process) |
| Improving telecommunication channels for data transmission *Having reliable communication channels will ensure that surveillance activities are well coordinated* *Unreliable communication channels lead to delays in follow-ups on surveillance data* *Adopting electronic communication channels at the lower levels will improve reporting* | Accurate case registration and reporting Enhanced resource capacity to support surveillance activities | √ √ | √ √ | √ √ | - | Yes | Framework component (Process) Components interlinkage |
| Improved means of transportation to facilitate surveillance activities *Unreliable transportation leads to delays in sending specimens to higher levels* *Transportation challenges hinder supervisory activities* | Enhanced resource capacity to support surveillance activities | √ | √ | √ | - | Yes | Framework component (Process) |
| Process IV: Strengthening surveillance system attributes | | | | | | | |

*(Continued)*

**Table 1.** (Continued)

| Phase II: Primary data studies Logical framework components *(Paraphrased statements)* | Component III: Output themes | Component IV: Outcome themes | | | Phase I: Systematic literature review themes | Phase III: Feasible actions (Yes/No) | Phase IV: Reviewed conceptual frameworks themes |
|---|---|---|---|---|---|---|---|
| Simplification of existing guidelines for completing reporting forms is required<br>*Training of health workers is required on application of the guidelines and reporting tools* | Accurate case registration and reporting Improved perception to surveillance system | √ √ | √ √ | √ √ | Improved reporting Simplicity | Yes | Framework component (Process) |
| Simplifying available forms to ease reporting of PC-NTDs<br>*Need for surveillance tools that are elaborate and easily understood by health workers to encourage reporting*<br>*Lower levels lack simplified tools to motivate reporting*<br>*Reporting of PC-NTDs could have been made easier if the reporting tools accommodated all the diseases* | Accurate case registration and reporting Improved perception to surveillance system | √ √ | √ √ | √ √ | Adequacy of reporting forms provision Simplicity | Yes | Framework component (Process) Components interlinkage |
| Simplifying PC-NTDs case definitions to ease application<br>*Majority of health workers in the lower levels lack knowledge on NTDs case definitions; making it difficult for them to report the diseases* | Improved perception to surveillance system Prioritisation of PC-NTDs surveillance activities | √ √ | √ √ | √ √ | Simplicity | Yes | Framework component (Process) |
| Simplifying methods for PC-NTDs surveillance data collection and analysis<br>*Data collection using the available tools is not efficient and easy to undertake; we require improved tools for reporting PC-NTDs* | Strengthened data analysis Improved perception to surveillance system | √ √ | √ √ | √ √ | Simplicity | Yes | Framework component (Process) |
| Health managers to support surveillance activities in the region<br>*Limited resource support for conducting surveillance activities influences health worker attitudes towards getting involved in surveillance activities*<br>*The system lacks acceptability among health workers due to lack adequate tools and resources to report and respond to NTDs* | Improved perception to surveillance system Prioritisation of PC-NTDs surveillance activities | √ √ | √ √ | √ √ | Acceptability | Yes | Framework component (Process) Stakeholders' involvement |
| Considering PC-NTDs to be of public health importance in the region<br>*NTDs are considered less important; this demotivates health worker involvement in surveillance activities concerning the diseases*<br>*PC-NTDs need to be prioritised from the highest to the lowest levels for effective surveillance to achieved* | Prioritisation of PC-NTDs surveillance activities | √ | √ | √ | Acceptability | Yes | Framework component (Process) Disease-specific perspective |

*(Continued)*

**Table 1.** (Continued)

| Phase II: Primary data studies Logical framework components *(Paraphrased statements)* | Component III: Output themes | Component IV: Outcome themes | | | Phase I: Systematic literature review themes | Phase III: Feasible actions (Yes/No) | Phase IV: Reviewed conceptual frameworks themes |
|---|---|---|---|---|---|---|---|
| Addressing challenges facing PC-NTDs surveillance activities with minimal delays *Health management need to respond promptly to challenges faced while undertaking surveillance activities* | Improved perception to surveillance system Prioritisation of PC-NTDs surveillance activities | √ √ | √ √ | √ √ | Stability | Yes | Framework component (Process) Disease-specific perspective |
| Existing surveillance systems to be well adapted to reporting all PC-NTDs in the region *Surveillance system prioritise reporting of diseases considered to be notifiable; therefore, not flexible to efficiently report other conditions such as NTDs* | Improved perception to surveillance system Prioritisation of PC-NTDs surveillance activities | √ √ | √ √ | √ √ | - | Yes | Framework component (Process) Disease-specific perspective |
| Existing surveillance systems to adapt easily to changes in PC-NTDs information needs *The surveillance system does not easily adapt to the changing NTDs distribution patterns; we hardly receive any updates on the diseases* | Improved perception to surveillance system Prioritisation of PC-NTDs surveillance activities | √ √ | √ √ | √ √ | - | Yes | Framework component (Process) Disease-specific perspective |

*—Recommendation actions that lacked consensus among stakeholders on their feasibility for implementation

√—Denotes an existing link to the concerned theme category

-—Denotes lack of an existing link to the concerned theme category

health importance, addressing surveillance activity challenges with minimum delays, ensuring surveillance systems are well-adapted to reporting all PC-NTDs and easily adapt to changes in information needs.

Consequently, inputs and processes in the planning phase facilitate achieving the intended results with anticipated outputs comprising; accurate case registration and reporting of PC-NTDs, improved case confirmation capacity, strengthened data analysis, improved feedback on surveillance data, improved epidemic preparedness and response, enhanced supervision of surveillance activities, improved training coverage on surveillance activities, enhanced resource capacity to support surveillance activities, improved health worker perceptions towards the surveillance system and prioritisation of PC-NTDs surveillance activities. Resultantly, the outcomes were thematically categorised into three sub-domains, which were linked to the likelihood of achieving either short, medium or long-term outcomes. The short-term outcome alluded to efficient transmission of surveillance data within the existing surveillance system while the medium-term outcome would result to improved surveillance data quality and lastly, the long-term outcome would be early detection and effective response action towards PC-NTDs. The overall impact of the resulting outcomes would be accurate estimation of disease burden, effective identification of disease transmission hotspots and implementation of targeted and cost-effective interventions.

The framework identified the intended results as a by-product of the input and process components making up the initial planned work phase (Fig 3). Consequently, the outputs were linked to one or more anticipated outcomes for strengthening PC-NTDs surveillance

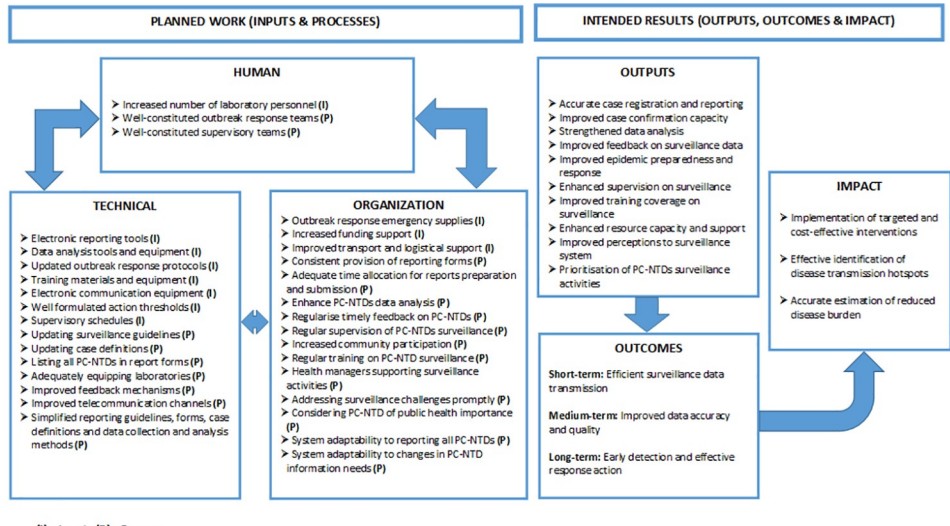

**Fig 3. Draft framework for improving PC-NTDs surveillance and response.**

and response (Fig 4). First, to achieve the short-term outcome regarding efficient surveillance data transmission and improved data accuracy required enhanced supervision of surveillance activities. Moreover, achieving the medium-term outcome relating to improved surveillance data accuracy and quality relied on strengthened data analysis, improved feedback on surveillance data and enhanced support supervision. Lastly, attaining the long-term outcome of early disease detection and effective response action relied on improved case confirmation capacity, reinforced data analysis and improved epidemic preparedness and response action. Overall, all the three anticipated outcomes–short-term, medium-term and long-term–relied on accurate

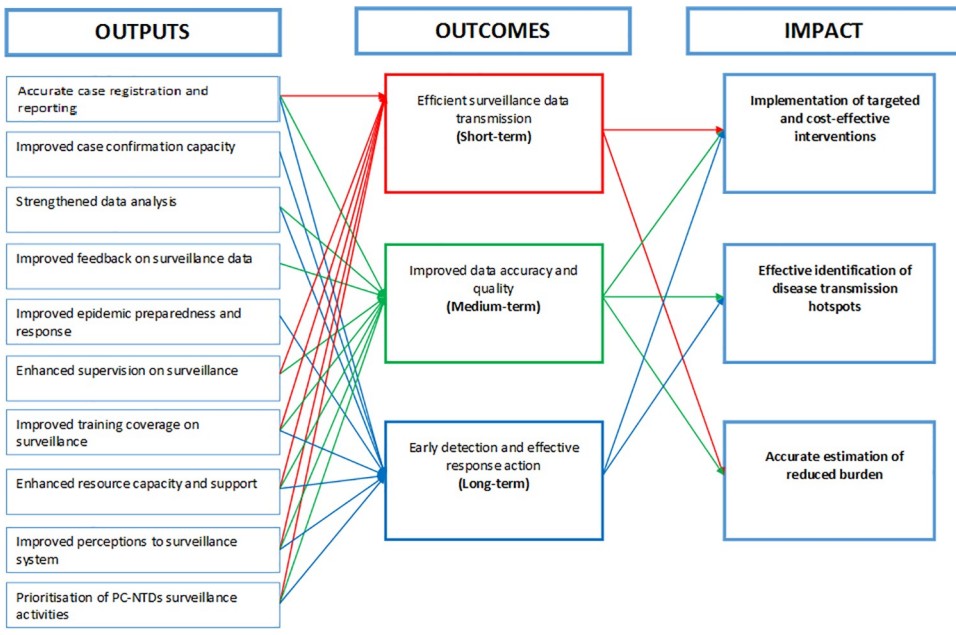

**Fig 4. Draft schematic illustration of component interlinkages in the intended results phase.**

case registration and reporting, improved training coverage on surveillance, enhanced resource provision, improved health worker perceptions towards the surveillance system and prioritising PC-NTDs surveillance activities. The proposed framework further linked the combination of efficient surveillance data transmission and improved data accuracy and quality to the impact component on accurate estimation of reduced disease burden. Additionally, improved surveillance data accuracy and quality, early disease detection and effective response action were linked to effective identification of disease transmission hotspots. Lastly, the framework identified a link between a combination of short and medium term outcomes to implementation of targeted and cost-effective interventions as an impact component.

## Framework validation

The purpose of framework development was to propose a logical approach for strengthening specific surveillance functions at the sub-national level to improve PC-NTDs surveillance and response. Therefore, a multi-phased approach was utilised to validate the proposed framework using: (i) consultative meetings with stakeholders at the sub-national and national levels and (ii) presentation of the draft framework in a conference meeting constituted of NTD researchers and policy experts. The aim of the consultative discussions was to review the draft framework components based on the expertise and experiences of healthcare stakeholders and decision makers. In addition, presentation of the draft framework in a scientific forum was intended to obtain further inputs to improve the framework components and assess scalability and adoptability of the framework especially at the sub-national level based on NTD research experts' opinions. The validation process presented an opportunity to ascertain the accuracy of information underpinning the draft framework and identified key points for framework refinement. The process allowed participants to comment freely on the framework components in an open forum, which allowed for extensive discussions and drew out diverse stakeholder and expert opinions.

## Validation process for the proposed framework

**Phase I.**   The stakeholders' consultative meetings at the sub-national level were conducted in two NTD endemic counties in Kenya. These counties included Baringo and West Pokot, which were purposely selected based on high prevalence of at least three co-endemic PC-NTDs in the regions. Fifteen participants from Baringo (9) and West Pokot (6) counties were enrolled in the validation exercise with representation from both county and sub-county levels. The participants comprised of directors of health, disease surveillance coordinators, epidemiologists, health information and records officers, NTD coordinators, public health officers and other key sub-national health stakeholders. In addition, stakeholder consultations ensued with relevant stakeholders drawn from the national disease surveillance unit (4) and the national NTD programme (3) in Kenya.

The consultative sessions first involved a larger group of participants, whereby the researcher provided a brief background to the framework and described the framework development process. In addition, participants were briefed on the core aim of developing the framework, framework components and their interlinkage. Participants were provided with copies of the draft framework, framework development process including the framework component matrix (Table 1) and a detailed illustration of framework components interlinkage. Subsequently, small group discussions of 3–5 participants were formed to further discuss and determine: (1) if the components of the framework required improvements or changes (2) whether the framework presented a logical flow of ideas and (3) the practicability of adopting the framework considering surveillance system capacity and accessible resources at the sub-

**Table 2. Framework validation resolutions.**

| Participants | Number of participants | Validation session output |
|---|---|---|
| *Baringo County (face-to-face and partly virtual)* | | - Distinct thematic categories to be identified in line with human, technical and organisational components to merge common activities of the planned work phase of the framework. |
| County Director of Health | 1 | |
| County Epidemiologist | 1 | - The framework needed to illustrate how "PC-NTD Elimination" is linked to the overall impact components. Participants suggested that the impact components of the framework |
| County Disease Surveillance Coordinator | 1 | should have a clear link to achieving the ultimate goal of disease elimination. |
| County Public Health Officer | 1 | - Stakeholders identified the need for indicating the target interventions to illustrate the |
| County Health Systems Manager | 1 | logical link between the specific interventions and framework components. |
| County Monitoring and Evaluation Officer | 1 | - Participants identified the need for reclassifying the outcome components to derive feasible |
| County Special Programmes Coordinator | 1 | short-term, medium-term and long-term outcomes. |
| Sub-county Disease Surveillance Coordinator | 1 | - Participants considered the framework to be an essential instrument to guide logical decision-making and recommended adoption at county level. |
| Sub-county Health Records and Information Officer | 1 | |
| *West Pokot County (face-to-face and partly virtual)* | | - Participants expressed the need for the framework to illustrate how the first phase of "planned work" led to the "intended results" phase using a directional arrow. |
| County Director of Health | 1 | - The framework needed to clearly illustrate how it is focused on strengthening the two core |
| County Disease Surveillance Coordinator | 1 | medical interventions for PC-NTDs. |
| County Public Health Officer | 1 | - Participants suggested the need to revise impact components to portray sustainable |
| County Health Records and Information Officer | 1 | achievements following from attaining the long-term outcomes. |
| Sub-county Disease Surveillance Coordinator | 1 | - Impact components needed to be linked to the key target of NTDs elimination–by ending epidemics resulting from the diseases–as outlined in the third Sustainable Development |
| Sub-county Health Records and Information Officer | 1 | Goal. <br> - Participants endorsed all framework components and supported adoption of the framework to guide decisions to improve PC-NTDs surveillance and response at the sub-national level. |
| *National level (face-to-face and partly virtual)* | | - Participants identified need for sub-national stakeholders to formulate detailed log frames to provide clear guidelines to achieving the desired outcomes and impact. |
| National disease surveillance unit | 4 | - The schematic illustration of intended results phase needed revision to illustrate the |
| National NTD programme | 3 | interlinkage between outcomes and impact components for clarity of the final phase. <br> - All participants supported adoption of the framework at the sub-national level to guide resource allocations and for use in prioritising implementation of activities to improve PC-NTDs surveillance and response. |
| *Royal Society of Tropical Medicine (RSTMH) Research in Progress Meeting (fully virtual)* | | - Participants indicated that the outcomes were to be interrelated with specific disease interventions. |
| Participants comprised of researchers, academicians and policy experts | >250 | - A common resolution among participants was that the desired outcomes and impact components needed rephrasing to depict sustainable efforts. <br> - Participants endorsed the framework as providing a logical approach to improving specific surveillance functions with a clear link to the targeted interventions. |

national level. Broader forums were later reconstituted, involving all participants to further deliberate and reach consensus on resulting outcomes that emanated from the smaller groups.

Resolutions from the consultative sessions were used to improve the draft framework to its final status. The concerned stakeholders reached a number of resolutions on specific framework components (Table 2). First, thematic sub-categories were established in the planned work phase of the framework. These sub-categories were in line with the broader themes of human, technical and organisational components. Specific sub-themes making up the inputs and processes included human resource management, data management, standards and guidelines, tools and equipment, communication, resource support, surveillance activities management and surveillance system attributes. Secondly, the short-term outcomes were revised to include efficient transmission of surveillance data and improved data accuracy. Medium-term outcomes involved early disease detection and response action and improved surveillance data quality. Lastly, long-term outcomes included; improved implementation of targeted and cost-effective interventions, enhanced identification of disease transmission hot-spots, and improved estimation of overall disease burden. Furthermore, long-term outcome

components were linked to target interventions. For instance, enhanced identification of disease transmission hotspots was a desired long-term outcome, which was linked to intensified CF. Thirdly, impact components were revised to signify sustainable efforts resulting from the long-term outcomes towards achieving the ultimate goal of PC-NTDs elimination. Therefore, the overall impact components were reviewed to include reduced costs for implementing PC interventions, halted disease transmission and reduced disease burden relating to PC-NTDs. Fourthly, a link was established between specific medical interventions associated with framework components. Therefore, the framework was shown to target improvement of PC and intensified disease management (IDM) interventions. Moreover, impact components of the framework were linked to the ultimate goal of disease elimination as outlined in the NTD roadmap and in achieving the third SDG [70,71]. Further resolutions alluded to illustrating how the "planned work" phase led on to the "intended results" phase using a directional arrow. Lastly, participants at the sub-national level endorsed the framework as identifying feasible actions for strengthening specific surveillance functions in relation to PC-NTDs.

On the other hand, framework validation at national level included a schematic illustration of the intended results phase focusing on the interlinkage between desired outcomes and impact components. Moreover, the logical framework was supplemented with detailed log frames for the desired impacts to be realised. The log frames for achieving the three distinct impact components constituted objectively verifiable indicators, sources of information, inputs (resources) and relevant assumptions (S4, S5 and S6 Tables). Log frames formulated through consultations with concerned stakeholders depicted that the overall impact of disease elimination was achievable through an integrated approach. This meant consolidating various inputs and processes at the planned work phase to achieve the intended results and outcomes. Essentially, stakeholders agreed that the framework offered a logical approach to improve PC-NTD surveillance and response from the initial planned work phase to achieving the intended results and the ultimate goal of disease elimination.

**Phase II.**    The framework development process was further presented in a virtual conference meeting in the form of a research poster. The forum was partly composed of breakout sessions that enabled extensive discussions of the framework components with various NTD researchers and policy experts. An interrelation was established between the desired outcomes and the concerned disease interventions, similar to stakeholder resolutions in the first phase. The interlinkage provided a clear link between implementing specific control interventions and achieving the overall goals. Additionally, impact components were rephrased to portray "sustained" control efforts even post-elimination. Lastly, participants agreed that the framework provided a logical approach to guide stakeholders' actions to achieve improved PC-NTDs surveillance and response in endemic settings (Fig 5). Stakeholder consultations in both phases aided in identifying critical interlinkages between the long-term outcomes and impact components in various combinations (Fig 6). First, improved implementation of targeted and cost-effective PC interventions, in addition to enhancing identification of disease transmission hotspots through active CF would have an overall impact of reducing costs for implementing treatment interventions. Secondly, identification of disease transmission hotspots and improved estimation of overall disease burden would inform effective implementation of PC interventions and intensify disease management, which would contribute to halted disease transmission. Finally, a combination of all long-term outcomes on implementation of targeted PC interventions, identification of disease transmission hotspots through active CF and improved estimation of overall disease burden to inform IDM would achieve reduced disease burden relating PC-NTDs. The overall impact of reduced costs for treatment interventions, halted disease transmission and reduced disease burden contribute towards achieving sustainable PC-NTDs elimination.

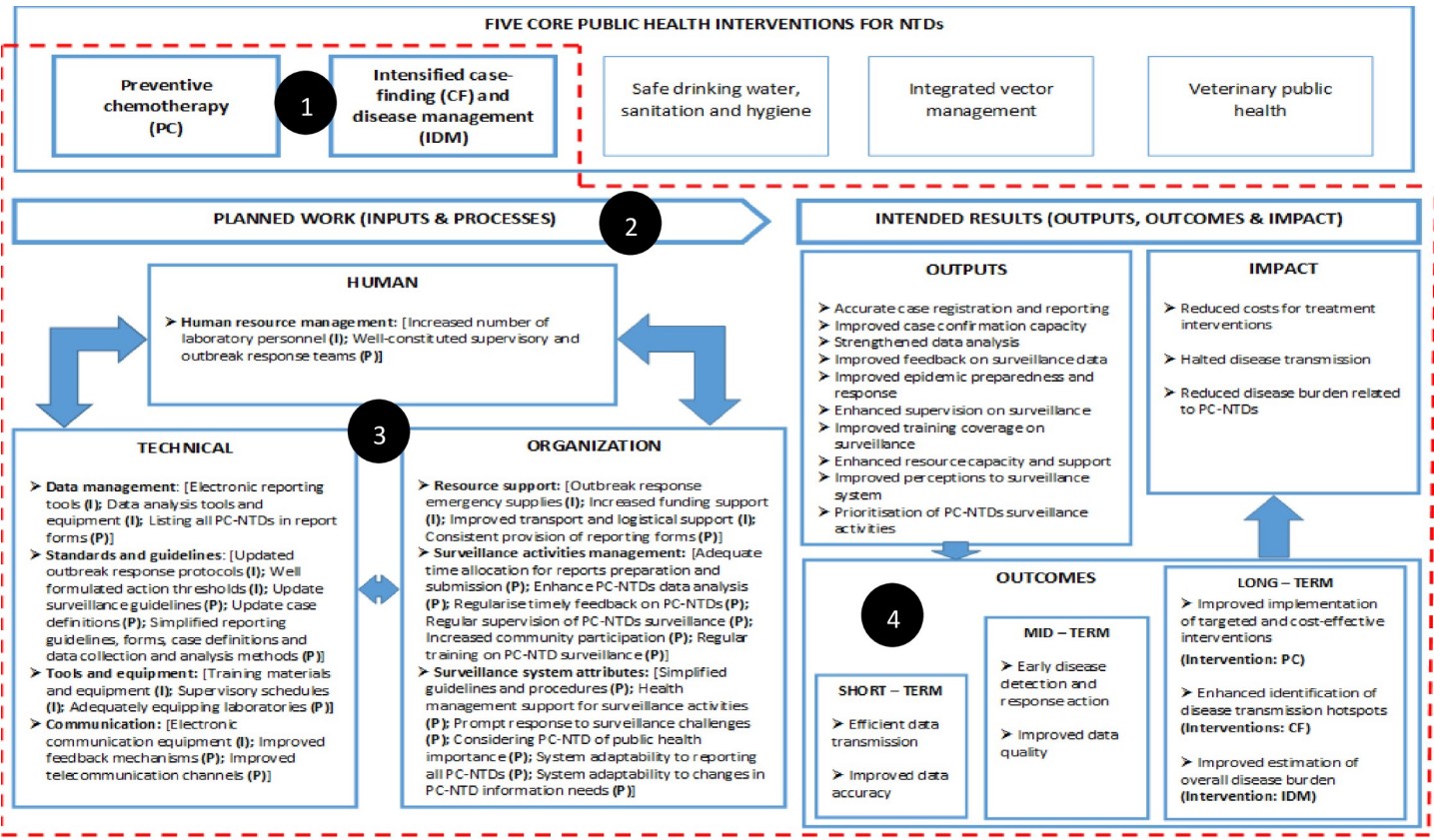

**Validation Phase Outcomes: (1)**: Link between the framework and the two core medical interventions; **(2)**: "Planned work phase" leading to the "intended results phase"; **(3)**: Human, technical and organization components themed into relevant sub-categories; **(4)**: Reclassification of outcome components (i.e. short-term, medium-term and long-term)

**Fig 5. Final validated framework for improving PC-NTDs surveillance and response.**

## Discussion

We developed a logical framework for guiding implementation of recommendations to improve surveillance and response to NTDs amenable to chemoprophylaxis. The framework was grounded on existing conceptual frameworks and adopted a stakeholder-oriented approach to determine the framework components and their interlinkage [21–23,67]. The feasible actions identified in the third phase of framework development were assessed and categorised as either constituting input or process components. The outputs, outcomes and impact components were thematically derived from reviewed literature in the first phase and analysis of participants' open-responses in the second and third phases. Recommendations drawn from Phases I, II and III were incorporated into a logical component matrix to assess interdependence between the inputs, processes, outputs, outcomes and overall impact. Furthermore, we utilised the W. K. Kellogg Basic Logic Model to derive key constructs and themes as described by concerned health stakeholders [66]. The logic model provides a systematic and visual course of action to recognise the link between resources or inputs, planned activities and the intended results [66].

Implementation of interventions involving multiple diseases is imperative and cost-effective especially in NTD endemic regions as opposed to parallel disease-specific strategies [72]. The WHO categories twenty diseases and conditions to constitute NTDs with two distinct

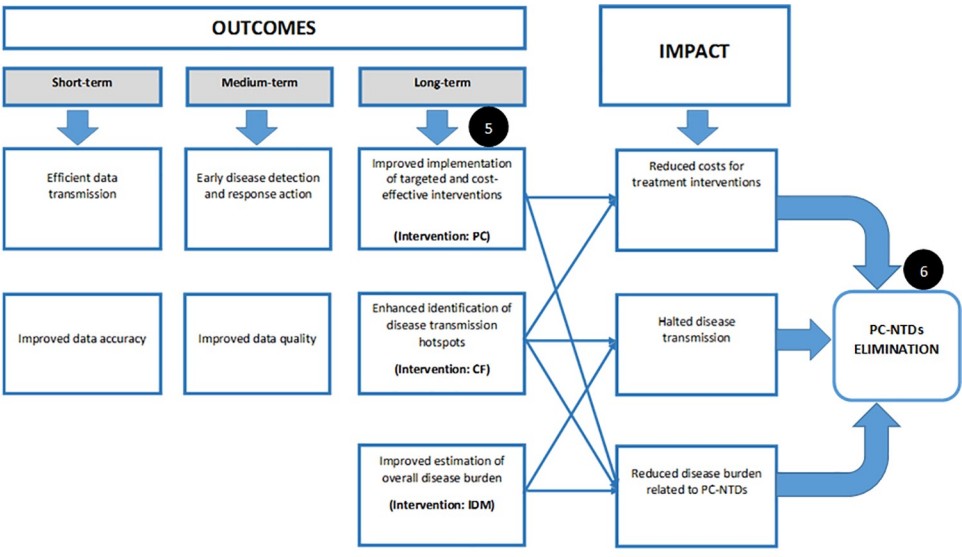

PC – Preventive Chemotherapy; PCNTDs – Preventive Chemotherapy Neglected Tropical Diseases; CF – Case Finding; IDM – Intensified Disease Management

Validation phase outcomes: (5): Details the link between long-term outcome components and the targeted interventions; (6): Impact components culminating to the ultimate goal of PC-NTDs elimination.

**Fig 6. Final schematic illustration of component interlinkages in the intended results phase.**

control measures; those diseases that are amenable to preventive chemoprophylaxis and those tackled through case management (CM) [7,9]. The focus of the current framework was on NTDs mainly controlled through PC interventions including lymphatic filariasis (LF), schistosomiasis, trachoma and soil transmitted helminth (STH) infections. Nonetheless, effective control of some of the PC-NTDs including LF and trachoma requires IDM [9,17]. Furthermore, WHO recommends five key strategies for prevention, control, elimination and eradication of NTDs [72,73]. These strategies include PC and IDM, which are the core medical interventions for curbing NTDs transmission. The other cross-cutting strategies include vector and intermediate host control, veterinary public health and provision of safe water, sanitation and hygiene [73]. Therefore, the present framework intended to link strengthened surveillance and response systems to the two core medical intervention strategies to achieve effective PC-NTDs control.

In the first phase of framework development, the systematic review revealed a dearth in surveillance assessment studies conducted in African countries with a focus on NTDs, which informed the second phase involving primary data surveys undertaken in an NTD endemic setting. An assessment of surveillance core, support and attribute functions regarding PC-NTDs targeted for control and elimination identified critical recommendations to improve their surveillance and response in Kenya. The third phase was aimed at assessing feasibility of implementing actions to improve PC-NTDs surveillance and response as perceived by healthcare workers in the second phase and generalised recommendations retrieved from the first phase. Consequently, the proposed framework components were informed by findings emerging from the three phases and a review of relevant conceptual frameworks, which formed the base for developing the current framework. Existing strategic plans focused on NTDs control and elimination formed the blueprint for framework development [29,30,73,74]. The framework looks to contribute towards efforts aimed at accelerating programmatic actions to control and eliminate NTDs as outlined in the recently launched Roadmap for NTDs 2021–2030 [70]. Essentially, the framework is inclined to improving the efficiency of medical interventions relating to implementation of PC, which is considered the safest and most effective intervention for controlling PC-NTDs. In addition, achieving NTDs-related morbidity reduction

requires IDM, which involves early disease detection through active CF, treatment and clinical management critical to NTDs control and elimination [9].

A critical input of the framework was the provision of updated surveillance guidelines, which corresponds to meeting NTD elimination targets by way of providing standardised guidelines essential for systematic collection of population-based data [75]. Furthermore, utilisation of WHO-approved guidelines aids disease mapping, monitoring and post-MDA surveillance approaches [75]. On the other hand, implementation of integrated and targeted PC interventions are largely impeded by the lack of complete data [76]. Hence, improving the existing reporting forms to ensure accurate and complete reporting of PC-NTDs is justified. There are no standardised methods for collecting PC-NTDs morbidity data except through baseline surveys during MDA campaigns [77]. However, morbidity data collected while concurrently conducting MDA usually lack quality and accuracy [78]. This affects effective implementation of NTD-targeted morbidity management strategies in endemic regions [77]. Furthermore, countries requiring validation and eventual certification as having eliminated specific NTDs need to accurately estimate the number of cases in endemic regions at the implementation unit level and monitor morbidity management and disability prevention (MMDP) [78]. This requirement corresponds to the framework's long-term outcome on improved estimation of overall PC-NTD burden.

A major challenge facing PC-NTDs control and elimination agenda in Kenya is minimal national and sub-national ownership of planned actions with most funding support for implementing interventions coming from development partners. In addition, limited involvement of local health stakeholders in NTDs control activities presents a pivotal sustainability challenge [30]. However, the validated framework components were identified based on convergence of opinion among various stakeholders. Stakeholder involvement and consensus is vital in guiding HIS strengthening and informing key healthcare decisions [20,21]. Analogous to the framework components, critical outcomes from the 'First Forum on Surveillance-Response System Leading to Tropical Disease Elimination' held on June 16–18, 2012 in Shanghai, identified key features influencing effective surveillance and elimination of NTDs [79]. These factors alluded to political will and good governance, commitment to utilise available resources to support disease elimination efforts, an established organisational and technical infrastructure with a competent health workforce, reliable healthcare services delivery and community involvement [79,80]. Therefore, critical elements of the validated framework are in line with meeting key targets to achieve NTDs elimination. Furthermore, increased knowledge on co-endemic NTDs may improve prompt case detection of multiple diseases that would eventually inform integrated management [17]. However, effective NTD training can only be achieved when disease-specific materials are utilised rather than adopting integrated health education approaches using complex guidelines [76]. This is consistent with the current framework process component on instituting regular training specific to PC-NTDs surveillance.

On the other hand, encouraging involvement of the community levels in surveillance activities is deemed to increase ownership and sustainability of the processes. This corresponds to efforts to empower communities living in NTD endemic regions through their participation in control interventions and formalizing the roles of community health workers [9,69]. Active CF at the peripheral level warrants reinforcement of community and health workers knowledge on NTDs through increased training and sensitisation [17]. Community health structures are an inherent part of health systems in developing countries especially regarding disease surveillance-related activities and efforts to achieve disease control. This is commensurate with the important assumption detailed in the log frames on increased community involvement through community-based surveillance activities [77]. Therefore, improved resource allocation at the sub-national level will support capacity building and training for community-based

health workers resulting to strengthened community-level structures [30]. Effective surveillance systems may contribute immensely to active detection of symptomatic cases at the lower levels. However, measurement of NTDs burden may be hindered in a majority of endemic settings with substantial proportions of asymptomatic cases similar to challenges facing malaria elimination programmes [81].

The framework's input and process components were broadly categorised into human, technical and organisational aspects. This was partly informed by the PRISM framework that clearly describes the interconnectedness between technical, behavioral and organisational factors [22]. Furthermore, the proposed framework adopted the HOT-fit concept that considers development of HIS according to an organisational plan of action while concurrently managing information technologies based on organisational needs [23]. The model considers human, organisation and technology as essential components of HIS, which influence information and service quality, system utilisation, user satisfaction, organisational environment and the net benefits [23]. Comparably, the validated framework combined human, technical and organisational input and process components to achieve the intended results. The framework further depicts a linkage between public health surveillance core, support and attribute functions and public health actions. This compares closely to the concept of linking public health surveillance activities to public health actions–acute or planned response–through a series of process-oriented measures to achieve the intended outputs and outcomes [19,20]. Achieving improved epidemic detection as a key output through processes involving data collection, analysis and epidemiologic investigations were synonymous with the framework for evaluating public health surveillance systems for early outbreak detection [36]. Likewise, ending NTD epidemics is a crucial target to achieving the third SDG [71].

Comparable to the framework, achieving NTDs elimination goals requires an intentional shift towards implementing PC strategies to not only attain conservative morbidity control but towards progressive disease transmission interruption [82]. Furthermore, implementation of cost-effective interventions rely on a shift from MDA to limiting the interventions to specific at-risk population groups [82]. Albeit the substantial progress so far achieved to control NTDs through cost-effective interventions, the remaining challenge of instituting adequate monitoring and surveillance efforts hinder sustained disease control and elimination [83]. Therefore, surveillance systems aid in the identification of disease transmission foci to inform targeted response actions in preference to mass interventional campaigns, which may otherwise result to misuse of scarce resources. In addition, continued mass treatment interventions may exert pressure on the disease causative agents resulting to drug resistance [84]. The validated framework further identifies the long-term component on improved estimation of overall disease burden as an enabler for IDM. For instance, effective surveillance systems could pinpoint regions with a high disease burden to inform appropriate actions [85].

Framework validation process identified opportunities for strengthening specific framework components. Stakeholder resolutions provided insights on the scalability, adoptability and feasibility of the framework to guide decisions for improving PC-NTDs surveillance and response at the sub-national level. Additionally, prior consensus statements in the Delphi survey identified actionable recommendations according to concerned health stakeholders, which informed components constituting the current framework. The log frames guided improved planning, implementation and management of activities to achieve desired goals [86]. They enabled structuring of framework elements–inputs, processes, outputs, outcomes and impact–to highlight the logical interlinkages [86]. Involvement of multiple stakeholders was required to assess the likelihood of implementing feasible actions. Therefore, stakeholders' resolutions determined the applicability and acceptability of such a framework at sub-national levels of PC-NTD endemic settings in Kenya.

## Strengths and limitations

A key strength of developing the framework lies within the third phase involving a Delphi survey, which identified the implementable actions considering health personnel perspectives. Therefore, the framework components were informed by feasible recommendations to improve surveillance and response to PC-NTDs at the sub-national levels. Validation of the framework involving relevant stakeholders was crucial and identified further opportunities for improving the framework components and assessing adoptability at the sub national level. However, to operationalise the framework, an in depth understanding of the decision making process at the sub-national level is vital to identify who makes decisions, what information is required to make decisions, and how and when decisions are to be made. Moreover, direct involvement of concerned communities will aid in validating the framework further to facilitate effective operationalisation. The framework postulates improving existing surveillance systems considering both an integrated and targeted process to controlling PC-NTDs. However, it may only apply to regions co-endemic of at least two or more PC-NTDs since the framework components intend to guide the decision-making process considering an integrated disease control approach. Furthermore, the framework validation phase was undertaken amid the COVID-19 pandemic, which resulted in conducting the validation process while strictly adhering to the Kenya Ministry of Health COVID-19 guidelines. This limited the number of face-to-face sessions undertaken and restricted the number of participants present within a given session. However, further resolutions from stakeholders were obtained through adopting alternative virtual mechanisms.

## Conclusion

Framework development was based on inclusion of factors that were initially assessed by key stakeholders regarding their importance and feasibility at the sub-national level. Further validation of the framework through a consultative process with concerned stakeholders aided in assessing its adoption, acceptability and utilisation for decision-making to improve PC-NTDs surveillance and response within the existing IDSR system. However, acceptability of novel surveillance and response approaches are dictated by the socio-cultural and political dynamics that should be considered when instituting control efforts within a specific unique setting [18]. The ultimate goal of the framework is to contribute to efficient implementation of two core medical interventions of the five interventional packages outlined in the WHO work plans for NTDs elimination in Africa [9,69]. The framework is inclined towards supporting implementation of interventions relating to PC for populations at risk of the infection and CF and IDM, which are critical to achieving PC-NTDs elimination. The framework will contribute towards sustainable control efforts by maintaining increased coverage of targeted interventions against NTDs to achieve the desired impact. Furthermore, attaining disease elimination and eventual eradication requires effective routine surveillance systems for early detection of possible instances of disease re-emergence both nationally and sub-nationally. However, health system strengthening frameworks in resource-constrained settings have predominantly focused on national levels with minimal focus on sub-national levels [87]. Therefore, the validated framework intends to guide decisions that will direct efforts towards assessing surveillance system performance at the sub-national level in view of PC-NTDs and inform implementation of relevant policies and control programmes. It would be pertinent for the WHO Regional Office for Africa in collaboration with national health ministries to adopt the proposed logical framework approach, which is tailored for developing countries settings to achieve improved performance of surveillance functions relating to PC-NTDs.

## Supporting information

**S1 File. Data collection tools.**
(PDF)

**S1 Table. Recommendations to improve PC-NTDs surveillance core activities.**
(DOCX)

**S2 Table. Recommendations to improve PC-NTDs surveillance support activities.**
(DOCX)

**S3 Table. Summary of existing conceptual frameworks.**
(DOCX)

**S4 Table. Log frame 1.**
(DOCX)

**S5 Table. Log frame 2.**
(DOCX)

**S6 Table. Log frame 3.**
(DOCX)

## Acknowledgments

The authors acknowledge the Kenya Ministry of Health and the County level authorities for granting permission to undertake the prior surveys that informed framework development. We also extend our sincere gratitude to all the healthcare personnel who participated in the surveys and framework validation process.

## Author Contributions

**Conceptualization:** Arthur K. S. Ng'etich, Kuku Voyi, Clifford M. Mutero.

**Data curation:** Arthur K. S. Ng'etich.

**Formal analysis:** Arthur K. S. Ng'etich.

**Investigation:** Arthur K. S. Ng'etich.

**Methodology:** Arthur K. S. Ng'etich.

**Supervision:** Kuku Voyi, Clifford M. Mutero.

**Validation:** Arthur K. S. Ng'etich.

**Visualization:** Arthur K. S. Ng'etich.

**Writing – original draft:** Arthur K. S. Ng'etich, Kuku Voyi, Clifford M. Mutero.

**Writing – review & editing:** Arthur K. S. Ng'etich, Kuku Voyi, Clifford M. Mutero.

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
