## [Decision Letter · Decision Letter 0]

22 Jun 2021

Dear Mr Ngetich,

Thank you very much for submitting your manuscript "Development and validation of a framework to improve neglected tropical diseases surveillance and response at the sub-national level in Kenya" for consideration at PLOS Neglected Tropical Diseases. As with all papers reviewed by the journal, your manuscript was reviewed by members of the editorial board and by several independent reviewers. In light of the reviews (below this email), we would like to invite the resubmission of a significantly-revised version that takes into account the reviewers' comments. 

We cannot make any decision about publication until we have seen the revised manuscript and your response to the reviewers' comments. Your revised manuscript is also likely to be sent to reviewers for further evaluation.

Sincerely,

Claudia Munoz-Zanzi

Associate Editor

Victoria Brookes

Deputy Editor

Reviewer's Responses to Questions

**Key Review Criteria Required for Acceptance?**

**Methods**

-Are the objectives of the study clearly articulated with a clear testable hypothesis stated?

-Is the study design appropriate to address the stated objectives?

-Is the population clearly described and appropriate for the hypothesis being tested?

-Is the sample size sufficient to ensure adequate power to address the hypothesis being tested?

-Were correct statistical analysis used to support conclusions?

-Are there concerns about ethical or regulatory requirements being met?

Reviewer #1: (No Response)

Reviewer #2: Is the PC-NTD framework design part of a major integrated disease surveillance and response -ISDR- program overhauling? Since the phase 1 methods are similar to those reported in reference 27.

The authors have as indicator of achievement of the framework Increased community participation in surveillance activities. As described, community members have been enrolled as healthcare workers for epidemiologic surveillance and they did participate in the framework´s survey phase. So, the absence of community-level representatives in the validation sessions of the framework at both Baringo and West Pokot counties (Table 2) is of some concern on ethical grounds, since they, the local communities, should be considered stakeholders in processes that affect them Can the functionaries at the validation sessions be consideren community-level stakeholders?

Reviewer #3: The methods section is detailed and well aligned to the goal of the study. It is adequate especially regarding the objectives of the study. It however, gets repeated almost totally in the results section and so a way of preventing this should have been devised in order to prevent this repetition.

**Results**

-Does the analysis presented match the analysis plan?

-Are the results clearly and completely presented?

-Are the figures (Tables, Images) of sufficient quality for clarity?

Reviewer #1: (No Response)

Reviewer #2: There is a extensive step by step description of the framework development. Table 1 is too long, but it seems unavoidable since it shows all variables or components of the framework that were taken into account.

Tre themes and subthemes of literature reviews in phase I are listed in detail in the text and it is tiresome. Instead, It should be presented as a table. 

The authors should avoid to repeat the same statement in the text and related tables as in table 2. 

Figures 2,3, 5 and 6 allow a clear under for the proposed framework process and interactions. But, Figure 4 is unnecessary, the information on it it should be placed in the text and integrated to respective figure legends ( 5 and 6)

Reviewer #3: The analysis as presented does match the analysis plan and clearly answers the research question according to the plan provided. The results though clear are too detailed, long and repetitive of earlier text in the manuscript. A means of summary the results to provide more focus and make it easier to read and follow should be explored.

The tables are of sufficient quality though they also provide to much information and details instead of being a summary of the findings. I would also suggest that the writers consider moving some of the less relevant tables to the appendix.

**Conclusions**

-Are the conclusions supported by the data presented?

-Are the limitations of analysis clearly described?

-Do the authors discuss how these data can be helpful to advance our understanding of the topic under study?

-Is public health relevance addressed?

Reviewer #1: (No Response)

Reviewer #2: The conclusion are well presented. 

However it as for most of public heath policies and interventions, sustainability is a key factor. The authors address that issue on the results but there is little or sparsely development in the conclusions.

Reviewer #3: The conclusions are supported by the data and unlike the methods and results sections the conclusions are more concise. The limitations of the study were discussed as well and the authors clearly indicated how the findings of the study would be used. However, the utility of the finding to WHO who provide technical guidance to the country programmes is not articulated in the paper.

**Editorial and Data Presentation Modifications?**

Reviewer #1: (No Response)

Reviewer #2: As described in the results. Also, the author introduce some acronyms that will requiere a complete wording for not experts as active CF (active case finding), HIS (Hospital information system) and so.

Reviewer #3: The manuscript is too long, a bit repetitive in presentation, and difficult to read. The introduction and the methods section are very detailed and most of the information and data under methods gets repeated in the results section instead of presenting a summary in the methods. The manuscript could benefit from

**Summary and General Comments**

Reviewer #1: (No Response)

Reviewer #2: The authors present a comprehensive four-phases methodology for revision and proposal for development and validation of a framework for PC-NTDs surveillance and response.

The manuscript presents a framework development process and a proposal that can be suitable as a blueprint for framework implementation in other countries with prevalent NTDs

Reviewer #3: The concept is perfect and focused on providing an M&E framework for NTDs which is a laudable goal and missing on many NTD programmes. The manuscript demonstrates a good understanding of M&E broadly and how it can be applied for NTDs M&E for best results and decision-making. Generally, the paper is highly informative, and the writers have done a great job but as indicated it is too long and makes it difficult to read and follow. I would suggest that the manuscript is broken up into more than one paper. Many of the tables could also be put in the appendix.

I suggest a major review of the paper before considering it for publication despite its utility.

Under the results section, the attributes of the surveillance functions need to be appropriately defined since they overlap in definition noting that the WHO, the organization which is responsible for providing technical norms and guidance has specific definitions for these attributes. An example is the attributes of data quality which includes accuracy, completeness, reliability, and timeliness and many of the other attributes are covered in meaning and definition under these attributes of quality data. The paper should therefore be able to define on its own the different expected data attributes to serve as guidance for the reader and at least should be provided in the introduction or literature search section.

I also expect that based on the literature review and results of the surveys there should be some prioritization of the recommendations and therefore indicators that could guide the development of a framework and also tools and guidelines.

Writers of this paper also suggest that NTDs are prone to outbreaks or epidemics which require epidemic or emergency preparedness and response, which is unusual since NTDs are generally known to be chronic conditions and have different data needs from other outbreak-prone conditions. Also, it was not discussed that PC-NTDs data collection are often one-off activities contrary to institutional data collection for conditions like malaria which involve routine data and facility-based data collection. PC-NTD data collection is also community-based with minimal and often insignificant facility-based data collection. These are important points that should have been discussed in the paper.

There also seems to be a mix in the presentation between data needs and operational issues and this needs to be clarified in the paper. The paper could also be better informed by a description of the health system for a better understanding of the M&E and surveillance needs at the different levels with a focus on the sub-national.

PLOS authors have the option to publish the peer review history of their article (what does this mean?). If published, this will include your full peer review and any attached files.

Reviewer #1: No

Reviewer #2: No

Reviewer #3: Yes: Nana-Kwadwo Biritwum
---

## [Decision Letter · Decision Letter 1]

17 Oct 2021

Dear Mr Ngetich,

We are pleased to inform you that your manuscript 'Development and validation of a framework to improve neglected tropical diseases surveillance and response at sub-national levels in Kenya' has been provisionally accepted for publication in PLOS Neglected Tropical Diseases.

Best regards,

Claudia Munoz-Zanzi

Associate Editor

Victoria Brookes

Deputy Editor

Reviewer's Responses to Questions

**Key Review Criteria Required for Acceptance?**

**Methods**

-Are the objectives of the study clearly articulated with a clear testable hypothesis stated?

-Is the study design appropriate to address the stated objectives?

-Is the population clearly described and appropriate for the hypothesis being tested?

-Is the sample size sufficient to ensure adequate power to address the hypothesis being tested?

-Were correct statistical analysis used to support conclusions?

-Are there concerns about ethical or regulatory requirements being met?

Reviewer #1: (No Response)

Reviewer #2: The manuscript meet the required criteria for acceptance. In a prior review, it was stated that the paper was valuable but it was long and repetitive. The authors were able to made some adjustment to make the manuscript more readable.

Reviewer #3: This is my second review of this paper and still think the methods is adequate and robust enough for answering the operational research questions under investigation leading results which can feed into the development of the proposed M&E framework.

**Results**

-Does the analysis presented match the analysis plan?

-Are the results clearly and completely presented?

-Are the figures (Tables, Images) of sufficient quality for clarity?

Reviewer #1: (No Response)

Reviewer #2: The results are well presented.

Reviewer #3: The analysis plan enables the researchers to deep dive into all the issues and responses relevant to the objectives of this paper. In my opinion it is clear and comprehensive for eliciting the required data and results.

**Conclusions**

-Are the conclusions supported by the data presented?

-Are the limitations of analysis clearly described?

-Do the authors discuss how these data can be helpful to advance our understanding of the topic under study?

-Is public health relevance addressed?

Reviewer #1: (No Response)

Reviewer #2: Discussion of results, conclusion and limitations for the design of the framework was consistent with the results and circumstances of the assessment and development.

Reviewer #3: The conclusions are clearly supported by the data collected and analysed. The utility of this paper towards informing WHO guidance for Neglected Tropical Diseases at the sub-national level and laudable and explicit in the paper.

**Editorial and Data Presentation Modifications?**

Reviewer #1: (No Response)

Reviewer #2: No suggestions

Reviewer #3: A description of the neglected tropical diseases programme implemented in the two districts especially for PC-NTDs in the beginning of the manuscript would provide a very good understanding of this well written paper. There is also an overemphasis on NTDs being prone to outbreaks and having the potential to be categorized as public health emergencies requiring requiring emergency preparedness, but knowing PC-NTDs to be mainly chronic condition, requires that these statements should be made with caution. Nevertheless, conditions like schistosomiasis may present as such. I still find the paper a bit too long with a lot of information in the text also repeated in the tables. This paper should be published with or without further modifications.

**Summary and General Comments**

Reviewer #1: All my earlier comments have been satisfactorily addressed, I have no further comments.

Reviewer #2: As commented earlier, the framework design can be used as blueprint for similar surveillance and response systems proposals in other regions affected by NTDs

Reviewer #3: A description of the neglected tropical diseases programme implemented in the two districts especially for PC-NTDs in the beginning of the manuscript would provide a very good understanding of this well written paper. There is also an overemphasis on NTDs being prone to outbreaks and having the potential to be categorized as public health emergencies requiring requiring emergency preparedness, but knowing PC-NTDs to be mainly chronic condition, requires that these statements should be made with caution. Nevertheless, conditions like schistosomiasis may present as such. I still find the paper a bit too long with a lot of information in the text also repeated in the tables. This paper should be published with or without further modifications.

PLOS authors have the option to publish the peer review history of their article (what does this mean?). If published, this will include your full peer review and any attached files.

Reviewer #1: **Yes: **Collins Okoyo

Reviewer #2: No

Reviewer #3: **Yes: **Dr. Nana-Kwadwo Biritwum

---

## [Editor Report · Acceptance letter]

22 Oct 2021

Dear Dr Ng'etich,

We are delighted to inform you that your manuscript, "Development and validation of a framework to improve neglected tropical diseases surveillance and response at sub-national levels in Kenya," has been formally accepted for publication in PLOS Neglected Tropical Diseases.

Best regards,

Shaden Kamhawi

co-Editor-in-Chief

Paul Brindley

co-Editor-in-Chief
